# Fatal intentional drowning in Australia: A systematic literature review of rates and risk factors

**Muthia Cenderadewi[1], Richard C. Franklin[1,2]\*, Amy E. Peden [1,2,3], Sue Devine[1]**

**1** College of Public Health, Medical and Veterinary Sciences, James Cook University, Townsville, Australia, **2** Royal Life Saving Society—Australia, Sydney, Australia, **3** School of Public Health and Community Medicine, University of New South Wales, Sydney, Australia

\* richard.franklin@jcu.edu.au

**Data Availability Statement:** This study reports the findings of a systematic literature review. All relevant data are in the supporting information files.

## Abstract

### Introduction

Unintentional drowning deaths are only part of the drowning profile, with little attention being paid to intentional drowning in Australia. Strategies for the prevention of intentional drowning deaths are likely to be different from unintentional. Quality documentation, analysis and dissemination of intentional deaths data is crucial for developing appropriate strategies for prevention.

### Objective

To conduct a systematic literature review to investigate the mortality rates and risk factors of intentional drowning deaths in Australia.

### Methods

A systematic search guided by PRISMA was performed using Ovid MEDLINE, CINAHL, PsycINFO (ProQuest), Scopus, Google Scholar, and BioMed Central databases to locate relevant original research articles published between 2007 and 2018.

### Results

Ten papers reporting the mortality rates and risk factors of intentional drowning deaths in Australia published between 2007 and 2018, with study periods of the included articles spanning from 1907 to 2012, were reviewed. Most studies investigated suicidal drowning deaths in Australia, none reported homicidal drowning deaths. The downward trend of fatal suicide drowning was identified in Australia. The annual rate of intentional drowning between 1994 and 2012 can be inferred from eight studies, ranging from 0.06 to 0.21 for nation-wide mortality rates. The highest annual state-wide mortality rate was identified in the state of Queensland, ranging from 0.02 to 0.11 per 100,000 individuals. Of four studies examining the risk factors of fatal intentional drowning in Australia, being of older age groups, being female, and the presence of substance use were identified as important

**Funding:** The author(s) received no specific funding for this work.

**Competing interests:** The authors have declared that no competing interests exist.

factors for suicidal drowning deaths. The national-scale proportion of suicide drowning in Australia, ranging from 2% to 3% of all intentional self-harm deaths, was also identified.

## Conclusion

Limited publications reporting the mortality rates and risk factors of intentional drowning deaths in Australia were identified. Being of older age groups and being female were recognised as factors for suicide drowning deaths, and psychoactive substances were widely identified amongst cases. Future research on improving death reporting systems and the legal framework for medico-legal death investigation, along with the investigation of the risk factors of intentional drowning, are required to inform the planning, implementation, and evaluation of prevention interventions for intentional drowning deaths in Australia.

## Introduction

Drowning is the third leading cause of unnatural deaths worldwide.[1–3] According to the World Health Organization (WHO) Global Report on Drowning, annually the lives of 320,000 people are lost due to drowning worldwide. [4] A recent study on the global, regional and national burden of unintentional drowning mortality from the Global Burden of Disease 2017 by Franklin et al, reported a global estimate of 295,210 fatalities was attributed to drowning in 2017.[1] However, the drowning statistics reported may underestimate the actual magnitude of all drowning deaths as has been suggested in several previous studies, particularly regarding unintentional drowning fatalities.[5–8] Peden et al [6] identified a 40% under-report between WHO methodology and unintentional fatal drowning in Australia. A similar study also estimated a 35% higher unintentional drowning mortality rate in Australia and the United States (US) when compared to WHO estimates, with other studies estimating a gap of 50%.[7] The recent study on the burden of unintentional drowning mortality from the Global Burden of Disease 2017, including in Australia, also did not include intentional drowning, which underlines the need for further study on intentional drowning deaths.[1]

Publications on the extent and trends of intentional drowning death are limited, even in high income countries where injury surveillance is better established than in developing nations.[9, 10] Most studies on intentional drowning explore suicides by drowning, with limited attention on submersion deaths due to assault.[9, 11–20] Despite the limited number of publications, a high proportion of intentional drowning deaths were identified amongst all fatal drowning incidents reported in previous studies.[21–26] A study in Sweden over the period of 1992–2009 [2], reported 31% of all drowning deaths in the country were identified as suicidal. In contrast to the disproportionately higher risk for unintentional drowning amongst male populations worldwide, more than half of suicide drowning cases were performed by females (55% versus 21%, p<0.001).[2, 27] The evidence on the proportion of intentional drowning of all suicidal deaths, however, shows varying proportions across the world. A study based in Milan, Italy, described suicidal drowning as the fifth highest cause of death due to self-harm registered in the region [11], while much lower proportions where identified in Norway (9%) [12], Canada (6%) [28], the US (1%) [15], and Taiwan (1%) [29].

As reported by the WHO in its 2012 publication 'Public Health Action Framework For The Prevention Of Suicide', death by suicide has been acknowledged as a major health issue throughout the world, with close to one million lives being lost due to suicide annually.[30] In

2015, suicide was the 10th and the 13th leading cause of death in the US and Australia, respectively. [31, 32] The trend of suicide deaths has also been consistently increasing, and the 2% contribution of suicide deaths to the global disease burden in 1998 is predicted to increase by one-third by 2020. The suicide mortality rate in Australia has also shown an upward trend, from 10 deaths per 100,000 individuals in 2006 to 14 deaths per 100,000 in 2015.[32] It is important to determine whether a similar trend is also observed for intentional suicide by drowning deaths in Australia. This will provide a better understanding of whether suicide drowning deaths have been contributing to the increasing trend of the suicide mortality rate in Australia, underlining the urgency of advocating for the enhancement of appropriate preventive measures for intentional drowning fatalities. This may differ from unintentional drowning prevention efforts.[32]

The link between suicidal drowning deaths and psychiatric disorders has also been reported in previous studies worldwide.[9, 33–36] A study in South Korea by Woo et al, [37] reported 21% of intentional drowning deaths between 2000 and 2012 were documented with a history of depressive episodes; although other risk factors, including being widowed and being out of work-force, also contributed as important factors among older age groups. Thus, in the context of suicide prevention in Australia, particularly with the high burden of mental health disorders in Australia [38–41] and the devastating impacts of intentional deaths [42–45], it is essential to provide reliable information on the trend and rates of intentional drowning. Such information will inform the development, implementation and evaluation of intentional drowning prevention and mental health promotion strategies in Australia.

Through a systematic review of the literature this study aimed to describe the epidemiology, the mortality rates and risk factors of intentional drowning deaths in Australia (i.e. both suicide and assault). This study answers the following questions:

- What is the mortality rate of intentional drowning in Australia?

- What proportion of suicide mechanisms is drowning in Australia?

- What are the risk factors for fatal intentional drowning in Australia?

## Materials and methods

### Search strategy

A systematic search was performed using six databases: 1) Ovid MEDLINE, 2) CINAHL, 3) PsycINFO (ProQuest), 4) Scopus, 5) Google Scholar, and 6) BioMed Central, to locate original research articles published between 2007 and 2018. The most exhaustive search strings (i.e. those that returned the highest number of results) were applied in the systematic search, in accordance to each database utilised (Table 1). The Preferred Reporting Items for Systematic

**Table 1. Database searching.**

| Databases | Search strings |
|---|---|
| Ovid MEDLINE | exp *Drowning/ AND (exp *Suicide/ OR exp *Homicide/ OR exp *Violence/) |
| CINAHL | (MH "Drowning") AND ((MM "Suicide+") OR (MM "Homicide+") OR (MM "Violence+")) |
| PsycINFO (ProQuest) | Drowning AND (MJSUB.EXACT.EXPLODE("Violence") OR MJSUB.EXACT.EXPLODE ("Suicide") OR MJSUB.EXACT.EXPLODE("Homicide")) |
| SCOPUS | (drowning) AND ((suicide OR homicide OR violence)) AND (LIMIT-TO (AFFILCOUNTRY, "Australia")) |
| Google Scholar | allintitle: drowning AND (suicide OR homicide OR violence) |
| BioMed Central | (drowning) AND ((suicide OR homicide OR violence)) AND australia |

Reviews and Meta-Analyses (PRISMA) flow diagram [46] was used to report the systematic literature review process. The PRISMA checklist for this study can be found in S1 File.

**Selection process.**   The selection process of all identified records was as follows:

**Title and abstract screening.**   The titles and abstracts of all records identified through database searching were screened by the inclusion and exclusion criteria (outlined in Table 2), to ensure the relevance of the studies included for the evidence-based review. If a study examined both intentional and unintentional drowning cases, it was included if intentional drowning rates and risk factors could be differentiated from the unintentional cases.

**Full-text screening.**   The full-text version of the identified articles were then appraised to examine the methodological quality of the studies, using the McMaster appraisal guideline for quantitative and qualitative studies. [47] The PRISMA flow diagram below summarises the selection process of this review (Fig 1).

## Data abstraction

The following data were abstracted from the studies: year of study and publication, data source, study design, scale of study, manner of intentional drowning deaths investigated, and mortality rates and risk factors of intentional drowning in Australia.

The nation-wide annual rates of intentional drowning were then inferred from studies that report nation-wide suicide drowning deaths and sourced their data from national scale sources. Such sources included the Australian Bureau of Statistics (ABS), the Australian Institute of Health and Welfare (AIHW), the National Coronial Information System (NCIS), or news reports. Rates were inferred by extracting the number of intentional drowning deaths from the papers in proportion to the population of the said year based on Australian population data provided by the ABS.

Meanwhile, the studies which only reported state-wide rates of intentional drowning deaths were not included in the nation-wide mortality rate calculations. Instead, these state-wide rates were inferred by extracting the number of intentional drowning deaths in the state from the paper in proportion to the population of the corresponding state for the said year based on the ABS data.

## Results

Electronic searches located a total of 574 articles from the six databases used, after duplicates were removed. Using the previously established inclusion and exclusion criteria (Table 2),

**Table 2. Inclusion and exclusion criteria.**

| Inclusion Criteria | Exclusion Criteria |
|---|---|
| • Published between 2007 and 2018<br>• Peer-reviewed journal articles<br>• Original research papers<br>• Observational studies<br>• Experimental studies<br>• Full-text available<br>• Published in English<br>• Drowning deaths in humans<br>• Specifically took place in Australia<br>• Investigating intentional drowning deaths<br>• Reported mortality rates and/or risk factors | • Non peer-reviewed journal articles, other types of publications<br>• Comprehensive scientific reviews, meta-analysis, statements of clinical standards, case reports, opinion pieces<br>• Only reporting unintentional drowning |

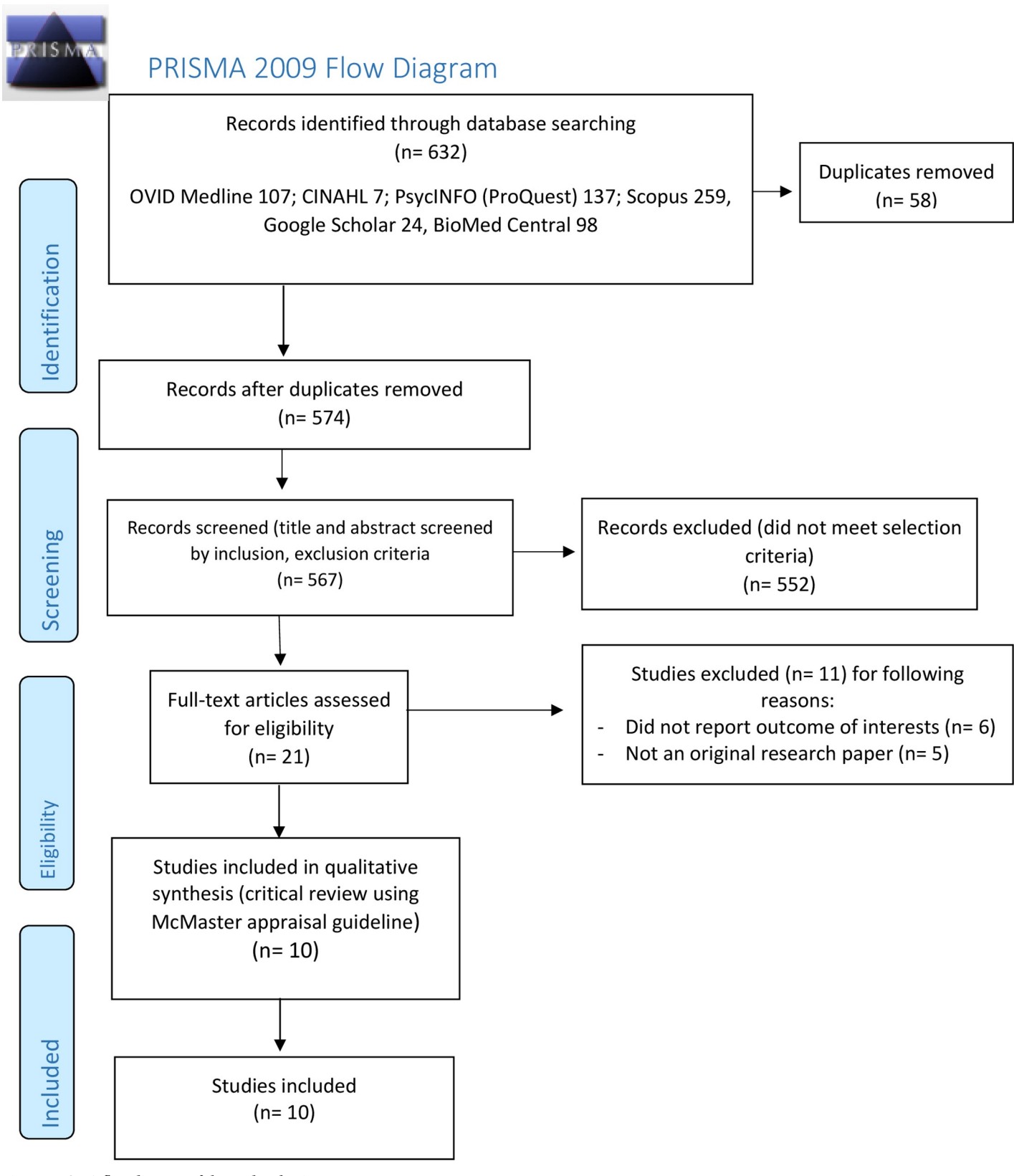

**Fig 1. PRISMA flow diagram of the study selection process.**

titles and abstracts of identified records were reviewed, resulting in 552 articles excluded due to ineligibility. After critically assessing the methodological quality and relevance of the full texts of the remaining articles using the McMaster appraisal form [47], 10 articles [48–57] were included for evidence-based review (Fig 1). Of ten articles included in this review, only one article [50] reported mortality rates of suicide drowning in Australia, while the majority of articles reviewed [48–50, 52–57] investigated suicidal drowning deaths in proportion to suicidal deaths of all causes in Australia. Amongst these, four articles [49, 50, 52, 53] also examined the risk factors of fatal intentional drowning in Australia. The full details of the data extraction, including any publication bias, can be found in S2 File.

Six articles [48, 50, 53–55, 57] were nation-wide studies, while the remaining four [49, 51, 52, 56] were of subnational scale, with two studies from the state of Queensland [52, 56] identified. All ten studies in this review utilised primary data sources. Four articles investigated national health and demographic statistics, provided by either the ABS or the AIHW [48, 50, 54, 58]. Four studies relied on suicidal drowning deaths data available in the NCIS [55–58] (Table 3). One study [54] used news reports as a data source, to determine the 'news-worthiness' of suicide deaths in Australia. Three studies [53, 54, 56] employed more than a single data source.

## Intentional drowning rates in Australia

Of the ten publications reviewed in this study, nine studies [48–50, 52–57] investigated intentional drowning deaths in Australia, compared to the proportion of suicidal deaths of all causes. One paper [51] determined the incidence of both intentional and unintentional drowning, as described in Table 4. However, none of the studies included in this review examined drowning deaths due to assault (Table 4).

Only one study [50] reported mortality rates of suicide drowning in Australia, highlighting the downward trend of intentional drowning as the preferred suicide method over a 90-year period of time (Table 4). Investigating the trend of mortality by suicide in Australia over the period 1907–1998, Donaldson, Bi and Hiller, 2007 [50] reported the trend of suicide drowning has been decreasing, from 3.10/100,000 in 1907 to 1.00/100,000 in 1998 among males, and from 1.25/100,000 in 1907 to 0.30/100,000 in 1998 in females.

The annual nation-wide rates of intentional drowning between 2000 and 2012, can be inferred from four studies [48, 53, 54, 55] (Table 5). Rates ranged from 0.06 to 0.21 for nation-wide mortality (Table 5). Meanwhile, the annual state-wide mortality rates between 1994 and 2012 can be inferred for three states: New South Wales [49], Victoria [51] and Queensland [53, 54, 55]. The lowest annual state-wide mortality rate was identified in the state of Victoria, with 0.01/100,000 from 1999 to 2011, and the highest observed in Queensland, ranging from 0.02 to 0.11 per 100,000 individuals, as described in Table 5. In comparison to all intentional self-harm deaths in Australia, the proportions of suicide drowning ranged from 2.1% to 2.7% [48,

**Table 3. Epidemiological data sources on intentional drowning mortality in Australia for studies included in this review.**

| Data Source | Number of studies |
|---|---|
| National health/demographic data (ABS, AIHW)[48, 50, 54, 58] | 4 |
| NICS[55–58] | 4 |
| State-level suicide registry[52, 56] | 2 |
| State-level ambulance registry[51] | 1 |
| Medico-legal/autopsy records[49] | 1 |
| News report[54] | 1 |

**Table 4. The epidemiology of intentional drowning deaths in Australia.**

| Authors | Year published | Year of study | Data source | Sample, setting | Scale of study | | Manner of death investigated | | Reported unintentional drowning rate/ rates | Proportion of suicide drowning among all suicide deaths |
|---|---|---|---|---|---|---|---|---|---|---|
| | | | | | National | Subnational | Suicidal | Homicidal | | |
| Byard R, Austin AE, Van Den Heuval C [48] | 2011 | 2008 | ABS | Nation-wide suicide deaths | ✓ | | ✓ | | N/A | 2.05% |
| Darke S, Duflou J, Torok M [49] | 2009 | 1997–2006 | Medico-legal/ autopsy records | Autopsied completed suicide deaths aged 15–60 years in NSW | | ✓ | ✓ | | N/A | 2.90% |
| Donaldson S, Bi P, Hiller JE [50] | 2007 | 1907–1998 | AIHW | Nation-wide suicide deaths | ✓ | | ✓ | | Yes (Male: 3.10/ 100,000 in 1907 to 1.00/100,000 in 1998; female: 1.25/ 100,000 in 1907 to 0.30/100,000 in 1998) | N/A |
| Dyson K, Morgans A, Bray J, Matthews B, Smith K [51] | 2013 | 1999–2011 | VACAR | All cases of out-of-hospital cardiac arrests due to drowning in Victoria | | ✓ | ✓ | | N/A | 0.89% |
| Koo YW, Kõlves K, De Leo D [52] | 2017 | 2000–2012 | QSR | Suicide cases classified as 'Beyond Reasonable Doubt' or 'Probable' in Queensland | | ✓ | ✓ | | N/A | 5.73% |
| Milner A, Witt K, Maheen H, LaMontagne A [53] | 2017 | 2001–2012 | ABS, NCIS | Nation-wide suicide deaths | ✓ | | ✓ | | N/A | 1.16% in males and 3.24% in females with occupations without access to lethal means of suicide |
| Pirkis J, Burgess P, Blood RW, Francis C [54] | 2007 | 2000–2001 | News reports, ABS | Nation-wide suicidal cases reported on sources of news | ✓ | | ✓ | | N/A | Over-reporting of suicide drowning (6.67% of all news report, in comparison to 1.67% of all suicides recorded by ABS) |
| Authors | Year published | Year of study | Data source | Sample, setting | National | Subnational | Suicidal | Homicidal | Reported unintentional drowning rate/ rates | Proportion of suicide drowning among all suicide deaths |
| Robinson J, Too LS, Pirkis J, Spittal MJ [55] | 2016 | 2010–2012 | NCIS | Nation-wide suicide deaths | ✓ | | ✓ | | N/A | 2.73% of all suicide deaths. Among adult suicides, cluster suicides were more likely to involve drowning (2.19% cluster cases versus 1.90% non-cluster cases). |

(*Continued*)

**Table 4.** (Continued)

| | | | | | | | | | | |
|---|---|---|---|---|---|---|---|---|---|---|
| Sveticic J, Too LS, De Leo D [56] | 2012 | 1994–2007 | QSR, NCIS | Suicide deaths in Queensland | | ✓ | ✓ | | N/A | Drowning was used significantly more frequently as the preferred suicide methods by reported missing persons than in non-missing victims (8.20% vs 1.80%; Σ2 = 39.53, df = 1, p<0.01). |
| Walter SJ, Bugeja L, Spittal MJ, Studdert DM [57] | 2012 | 2000–2007 | NCIS | Nation-wide (except for ACT and WA) deaths due to the external with discretionary or mandatory inquests. | ✓ | | | Undetermined drowning deaths | N/A | 15.37% of all discretionary inquests and 4.91% of all administrative<br><br>Investigations. Deaths by drowning, choking or suffocation (OR 5.12, 95% CI: 3.56–7.39) had much higher odds for discretionary inquest. |

ABS = the Australian Bureau of Statistics; AIHW = the Australian Institute of Health and Welfare; VACAR = the Victorian Ambulance Cardiac Arrest Registry; QSR = the Queensland Suicide Register; NCIS = the National Coroners Information System; NSW = New South Wales; ACT = Australia Capital Territory; WA = Western Australia; N/A = data unavailable.

55]. However, a higher proportion was reported in the state of Queensland with 5.7% suicidal drowning deaths among all suicide deaths documented in the state [52] (Table 5).

### Risk factors of intentional drowning in Australia

Of four articles [49, 50, 52, 53] examining the risk factors of fatal intentional drowning in Australia, socio-demographic characteristics, including age [52], gender [50], and substance use [53] were identified as important factors for suicidal drowning deaths. The highest incidence of suicidal drowning in Australia was observed in the older age groups of 65–74 and 75–84 years.[52] In fact, amongst older adults aged 65 years and over, the risk of intentional drowning in Australia significantly increased with age ($\chi$2 trend = 5.95, df = 1, p = 0.02), with individuals aged 65 to 74 years 0.65 (95% CI: 0.36–1.17) times less likely to perform suicide by drowning than individuals aged 75 to 84 years.[52] One study [50] investigated rates and trends of fatal suicide drowning in Australia by gender, between 1907 and 1998. The study found drowning was the most commonly used suicide method by females in the beginning of the last century, with a rate of 1.25 per 100,000 individuals in 1907.[50] Although this preference changed over time with the rise of non-violent suicide methods, such as poisoning, drowning was still commonly used by females in Australia in 1998 with an annual rate of 0.30/100,000. [50] On the contrary, drowning was the least common suicide method used by male victims in Australia between 1907 and 1998.[50]

The omnipresence of psychoactive substances among suicidal drowning cases in Australia was reported by Darke, Duflou [49]. This study revealed 91.7% of suicide drowning cases between 1997 and 2006 tested positive for substances, including alcohol (52.8%), pharmaceuticals (47.2%), other illicit drugs (19.4%), psychostimulants (8.3%), cannabis (5.6%), and opioids (5.6%).

**Table 5. Inferred annual national- and state-wide mortality rates of intentional drowning in Australia between 1994 and 2012.**

| Authors | Populations studied | Annual national- and state-wide mortality rate (per 100,000) | | | | | | | | | | | | | | | | | | |
|---|---|---|---|---|---|---|---|---|---|---|---|---|---|---|---|---|---|---|---|---|
| | | 1994 | 1995 | 1996 | 1997 | 1998 | 1999 | 2000 | 2001 | 2002 | 2003 | 2004 | 2005 | 2006 | 2007 | 2008 | 2009 | 2010 | 2011 | 2012 |
| Byard R, Austin AE, Van Den Heuval C[48] | Nation-wide suicide deaths | | | | | | | | | | | | | | | 0.21[a] | | | | |
| Milner A, Witt K, Maheen H, LaMontagne A [53] | Nation-wide suicide deaths | | | | | | | | 0.06[a] | | | | | | | | | | | |
| Pirkis J, Burgess P, Blood RW, Francis C[54] | Nation-wide suicidal cases reported on sources of news | | | | | | | 0.16[a] | | | | | | | | | | | | |
| Robinson J, Too LS, Pirkis J, Spittal MJ [55] | Nation-wide suicide deaths | | | | | | | | | | | | | | | | | 0.18[a] | | |
| Darke S, Duflou J, Torok M[49] | Autopsied completed suicide deaths aged 15–60 years in NSW | | | | 0.06[b] | | | | | | | | | | | | | | | |
| Dyson K, Morgans A, Bray J, Matthews B, Smith K[51] | All cases of out-of-hospital cardiac arrests due to drowning in Victoria | | | | | | 0.01[c] | | | | | | | | | | | | | |
| Koo YW, Kõlves K, De Leo D[52] | Suicide cases classified as 'Beyond Reasonable Doubt' or 'Probable' in Queensland | | | | | 0.11[d] | | | | | | | | | | | | | | |
| Sveticic J, Too LS, De Leo D [56] | Suicide deaths in Queensland | 0.02[d] | | | | | | | | | | | | | | | | | | |

[a]nation-wide

[b]New South Wales

[c]Victoria

[d]Queensland

## Discussion

This review aimed to describe the epidemiology, particularly the mortality rates and risk factors, of intentional drowning deaths in Australia. Ten publications on fatal intentional drowning in Australia were identified, with the majority of studies [48–50, 52–57] investigating suicidal drowning deaths in Australia, in proportion to suicidal deaths of all causes. No studies included drowning deaths due to assault, highlighting the lack of peer-reviewed publications on homicidal drowning deaths in Australia. This under-exploration of the epidemiology of intentional drowning deaths in Australia is similar to other developed nations, such as the US [15, 59, 60], the United Kingdom (UK) [14, 19, 61], Taiwan [17, 29, 62], Switzerland [63–66], Austria [63, 67, 68], and Germany.[66, 69] Studies from these countries also reported fatal suicidal drowning in proportion to suicidal deaths by all methods. Sweden, on the other hand,

has two comprehensive studies specifically investigating the rates and trends [2], and risk factors [9] of suicide drowning deaths, highlighting the importance of intentional drowning as a public health issue in the country. The dearth of epidemiological studies of intentional drowning deaths in high-income countries, including in Australia, underlines the gap between the more thoroughly understood epidemiology and advanced efforts of preventing unintentional drowning deaths, in comparison to the relatively under-exposed and under-represented intentional drowning deaths. This also highlights the possibility of further under-exploration of intentional drowning deaths in the resource-limited setting of developing nations, where injury surveillance systems, including for self-harm related events, might not be as well-developed as those available in high-income countries.[70, 71]

One study in this review [50], reported the trend of suicide drowning had been decreasing in Australia over the period 1907–1998. This downward trend of fatal suicide drowning is similar to several other high-income countries, for instance in Sweden [2] and Norway [12]. Thomas and Beech [14] also revealed the reduction of drowning as the preferred suicide method in England and Wales, from 23% of suicide deaths in 1901–1907 to 3% in 2001–2007, suggested restrictions to physical access, improvement in water transportation systems, and social preferences as factors influencing the changes in the popularity of some suicide methods. Although these data span a wide time period and are somewhat out of date now, this underlines the importance of modifying physical and social environments and developing mental health promotion to further reduce the risk for suicide drowning in the community.

This systematic literature review identified the national-scale proportion of suicide drowning in Australia ranging from 2% to 3% of all intentional self-harm deaths [48, 55]. These proportions of suicide drowning in Australia are comparatively lower than proportions observed in several other high-income countries including Germany [69], Sweden [72], and Switzerland [64]. It is also lower than the 16 European countries participating in the European Alliance Against Depression (EAAD) (Table 6).

There is a dearth of publications explaining the discrepancy between the lower proportion of suicide drowning among all suicide deaths in Australia in comparison to other developed nations. It is noted that there is a need for further research addressing the preference for suicide methods by countries, to inform the development of intervention strategies appropriate for each country, including Australia.

## Risk factors of fatal suicide drowning in Australia

This review reveals the under-exploration of risk factors in relevance to the incidence of suicidal drowning deaths in Australia. This poses an obstacle to the development, planning, and implementation of intervention strategies for intentional drowning in Australia.

**Fatal suicide drowning amongst elderly Australians.** Only one study in this review investigated age as a risk factor for intentional drowning deaths.[52] The risk of intentional drowning significantly increased with age ($\chi^2$ trend = 5.95, df = 1, p = 0.02), with the highest

**Table 6. Comparison of suicide drowning in Australia to other high-income countries.**

| Country | Year of study | Proportion of suicide drowning among all suicide deaths (%) |
|---|---|---|
| Australia [48, 55] | 1997–2006 | 2.05 |
| | 2010–2012 | 2.73 |
| Germany [69] | 1991–2002 | 3.77 |
| Sweden [72] | 1994–2009 | 6.79 |
| Switzerland [64] | 1969–2005 | 8.61 |
| European countries [66] | 2000–2004/2005 | 4.23 |

incidence of suicidal drowning observed in the older age groups of 65–74 and 75–84 years. [52] This is similar to findings of a previous study in Japan [20] and studies in England and Wales [13, 19, 61], where suicidal drowning was significantly higher in the older age group of 65 years or older than in individuals aged less than 65 years. Purandare et al, 2009 [61] contended that the choice of 'less-violent' suicide methods by older populations may be partially explained by aging-related cognitive deficits, potentially hindering the arrangement and enactment of the suicide act itself.

**Suicide drowning deaths in Australia by gender.** From the limited available data, it appears that there is a higher proportion of intentional drowning deaths among females than males in Australia. Donaldson, Bi and Hiller, 2007 [50] reported drowning as one of the most preferred suicide methods among female Australians across an extensive period between 1907 and 1998, while it was the least common suicide method used by male victims throughout the study period. Higher proportions of female suicides by drowning, in comparison to proportions in male suicide deaths, were also reported in Croatia [73], UK [13], Norway [12], and Japan [18].

In comparison to unintentional drowning in Australia, the finding of higher proportion of intentional drowning deaths in females in this review is also distinct from the epidemiology of unintentional drowning deaths in Australia. [74] As reported by Franklin, Scarr and Pearn, 2010 [74], 77% of all unintentional drowning cases in Australia between 2002 and 2006 were in males, as well as 80% of unintentional fatal drowning in Australian rivers [75].

**Psychoactive substances and fatal suicide drowning in Australia.** In this review, one study [49] identified psychoactive substances as a risk factor for intentional drowning and revealed the wide presence of psychoactive substances amongst suicidal drowning cases in Australia. Darke, Duflou and Torok, 2009 [49] reported those who died from suicide drowning in Australia were 2.56 times (95% CI: 1.24–5.29) more likely to be detected with benzodiazepines, and had 3.23 times (95% CI: 1.56–6.69) higher likelihood to be detected with antidepressants. This finding of the higher odds for benzodiazepines and antidepressants use amongst suicidal drowning victims is consistent with the report of the potential over-prescribing of benzodiazepine substances and antidepressants across Australia over the period of 1992 to 2011.[76–81]

**The under-exploration of psychiatric disorders among suicidal drowning victims in Australia.** None of the studies investigated in this current review explored the history of mental health disorders and previous suicide attempts amongst intentional drowning victims in Australia. This is despite the relevance of suicidal drowning deaths and psychiatric disorders identified in previous studies worldwide.[9, 33–36] The high proportion of prior psychiatric admissions and suicide attempts amongst suicidal drowning victims was reported in Sweden, with half of suicide drowning victims having had a history of hospitalisations due to mental health conditions. [9] One-third were released or left hospital less than one week before the suicide, and one-third had a history of previous unsuccessful suicide attempts.[9]

It is relevant to acknowledge that this current review identified suicide by drowning as the preferred method of suicide in only 2% to 3% of all intentional self-harm deaths in Australia. This highlights the importance of early recognition of mental disorders and previous suicide attempts, as well as the enhancement of mental health promotion and deliberate self-harm prevention, as a part of intentional self-harm and suicide prevention strategies, including for suicide by drowning.[9, 16, 18, 29, 64, 71, 73, 82]

## Recommendations for future research

This review has highlighted several opportunities for future research, with a particular focus on improving data collection systems for intentional drowning deaths and investigating risk

factors of intentional drowning. Such research will inform the planning, implementation, and evaluation of intentional drowning prevention.[4, 83, 84]

**Improving data collection systems.** There are challenges in investigating intentional drowning deaths, such as: (1) the difficulty in determining the intent of death, related to the legal framework and medico-legal investigation required to determine the manner of unnatural deaths, (2) the availability of a functioning national death registration system that records the causes of and circumstances surrounding the event of death, and (3) the availability of coding system with an appropriate level of detail for additional causes of death and co-morbidities to enable the investigation of the risk factors of intentional drowning death. [2, 85–89] These factors, as outlined in Fig 2, need to be strengthened in order to improve intentional drowning data collection, analysis, and dissemination to inform prevention strategies.

**Investigating risk factors of intentional drowning.** Another gap identified in this review is the under-exploration of risk factors of intentional drowning deaths. This review identifies only age [52], gender [50], and substance use [90] as factors for suicidal drowning deaths, highlighting the lack of investigation in risk factors for fatal intentional drowning in Australia. A number of risk factors have been suggested by several studies as being important factors for suicide attempts, including: 1) socio-demographic characteristics, including age [13], gender [66, 91], socioeconomic status [92], unemployment [92, 93], rurality [94], and indigenous identification [95, 96]; 2) substance dependence [91]; 3) psycho-pathology, including depressive disorders [91, 93], psychotic disorders [91], neurotic disorders [91], and previous intentional self-harm attempts [9, 33–36]; and 4) social dysfunction [91], including history of childhood abuse [97], social isolation [98], and homelessness [99]. A better understanding on the association of fatal intentional drowning and these risk factors, as illustrated in Fig 3, will also contribute to the development of public policy and the enhancement of community action in creating a supportive physical and social environment for mental health and intentional drowning prevention promotion in Australia. This conjunction of the enhancement of mental health promotion and deliberate self-harm prevention most likely differs from the development of unintentional drowning prevention strategies.

## Strengths and limitations

This review has met its aims, including providing analysis on the rates, trends, and risk factors of intentional drowning in Australia, by performing a systematic literature review to identify original papers published between 2007 and 2017. This study has also shown the consistency of the source of data, with 80% of the articles identified in this study using the national-scale data from the ABS, AIHW, and NCIS. This identifies the potential utilisation of the national-scale data, with some improvements in the data collection system as discussed in the Recommendations for Future Research section, for further investigation of intentional drowning death in Australia. Similar proportions of suicide by drowning were also noted between the included studies with national-scale sources of data [48, 49, 55], with drowning identified as the preferred method of suicide in 2% to 3% from of all intentional self-harm deaths in Australia.

However, several major limitations were identified in this review. First of all, it is possible that not all studies exploring intentional drowning death in Australia were identified and included in this review. In addition, data collection and reporting between articles hinders the comparison of age-specific, gender-specific, intentional drowning rates between studies reviewed. Similarly, differences in the definition of intentional drowning may exist between studies, further complicating comparison.

It is also noted in this current study that although only observational studies were identified in the systematic search, different study designs may have affected the heterogeneity and the

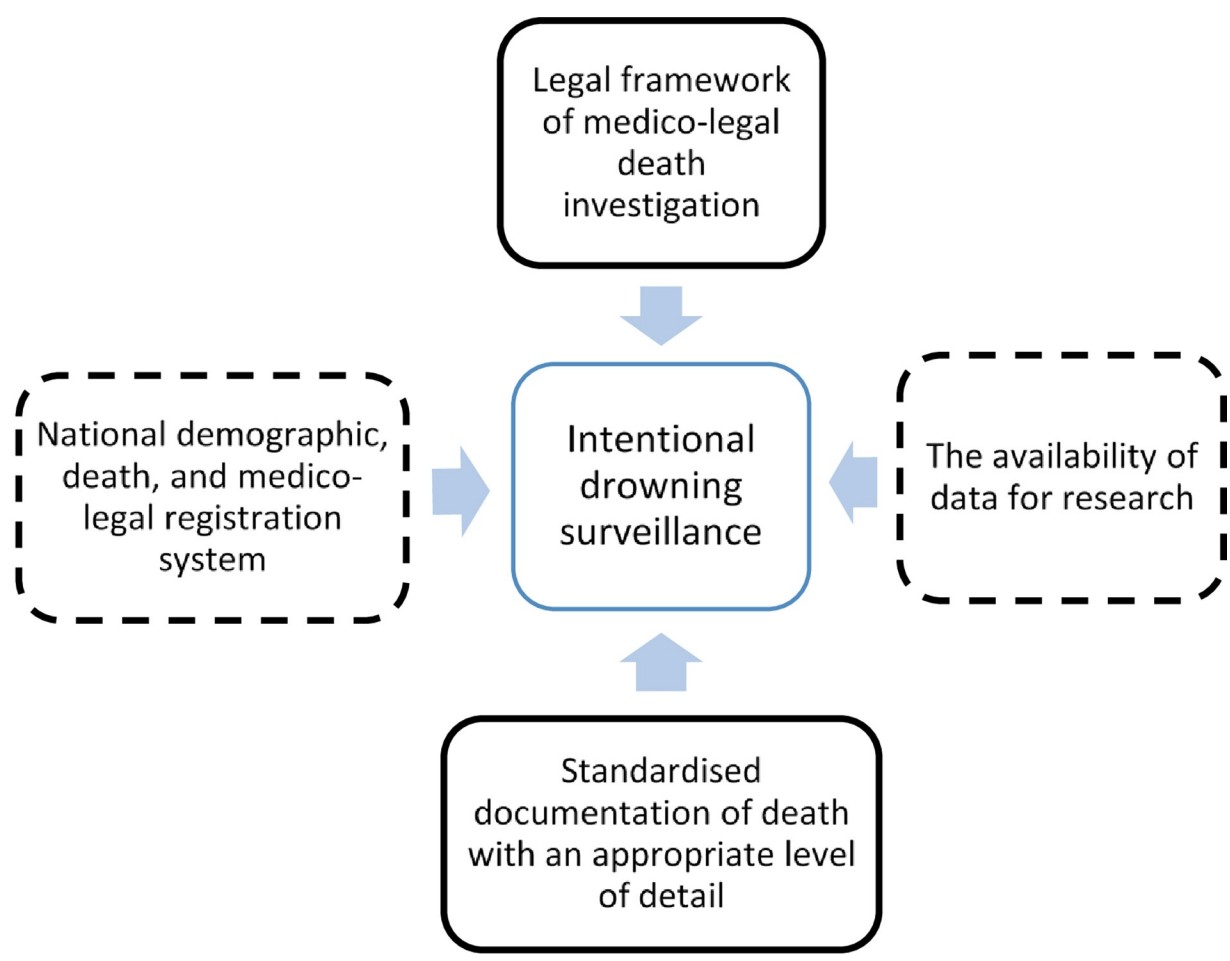

Note (in relevance to future research in Australia):

—— Needs to be explored

– – Sufficient

**Fig 2. Future research needed for improving intentional drowning data collection systems.**

potential for reporting bias. Moreover, the exclusion of grey literature, including government reports, policy statements, issues papers, and theses, may also exclude other sources of data on intentional drowning in Australia. The current study also highlights the need for more up-to-date research on intentional drowning in Australia, with the bulk of the studies included in this review 10 years old or older. Furthermore, no studies identified in this current study investigated drowning deaths due to assault, highlighting the lack of peer-reviewed publications on homicidal drowning deaths in Australia and the need to further research on this topic in the future.

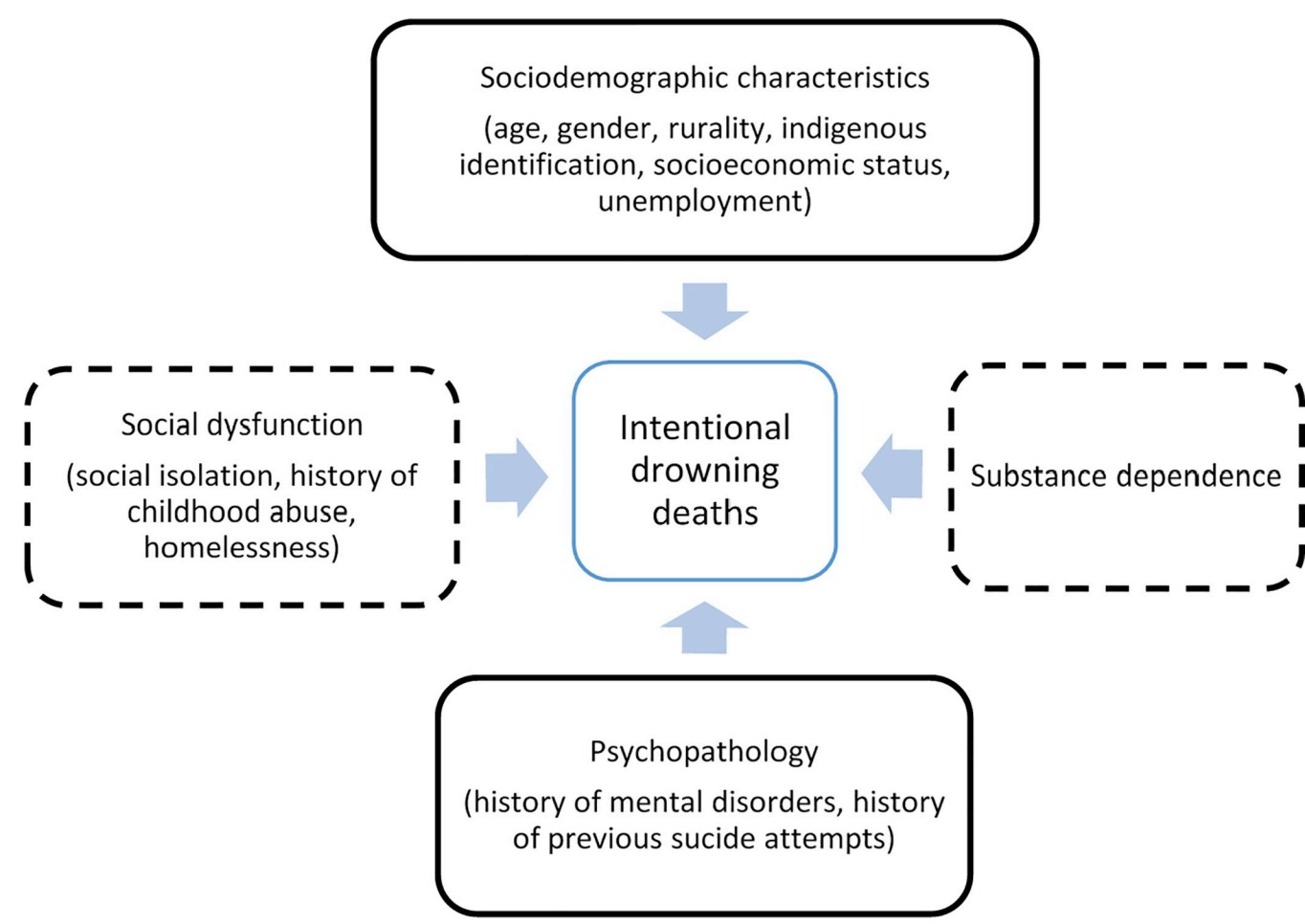

**Fig 3. Risk factors of intentional drowning deaths needed to be investigated in future research.**

## Conclusions

Limited publications reporting mortality rates and risk factors of intentional drowning deaths in Australia were identified in this review. A downward trend of suicidal drowning deaths was observed in Australia in one study, and in this study, being of older age groups was recognised as a risk factor of suicide drowning deaths in Australia. Psychoactive substances, particularly benzodiazepines and antidepressants, were identified amongst most suicidal drowning victims in the one study where this was investigated. This review identified suicide by drowning as the preferred method of suicide in only 2% to 3% of all intentional self-harm deaths in Australia. Improving death reporting systems and the legal framework for medico-legal death investigation, along with further research investigating the risk factors of intentional drowning, will

inform the planning, implementation, and evaluation of prevention interventions for intentional drowning deaths in Australia.

## Supporting information

**S1 File. PRISMA checklist.**
(DOC)

**S2 File. Data extraction table providing in-depth analysis of the included papers, including publication bias.**
(XLSX)

## Acknowledgments

The authors thankfully acknowledge the College of Public Health, Medical and Veterinary Sciences, James Cook University, for providing a supportive research environment. The publication of this paper is supported by James Cook University, Australia.

## Author Contributions

**Conceptualization:** Muthia Cenderadewi, Richard C. Franklin.

**Formal analysis:** Muthia Cenderadewi, Richard C. Franklin.

**Investigation:** Muthia Cenderadewi, Richard C. Franklin.

**Methodology:** Muthia Cenderadewi, Richard C. Franklin.

**Project administration:** Richard C. Franklin, Amy E. Peden, Sue Devine.

**Supervision:** Richard C. Franklin.

**Validation:** Richard C. Franklin.

**Writing – original draft:** Muthia Cenderadewi, Richard C. Franklin.

**Writing – review & editing:** Muthia Cenderadewi, Richard C. Franklin, Amy E. Peden, Sue Devine.

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
