## [Decision Letter · Decision Letter 0]

29 Oct 2019

PONE-D-19-21766

Fatal intentional drowning in Australia: A systematic literature review of rates and risk factors

PLOS ONE

Dear Associate Prof Franklin,

Thank you for submitting your manuscript to PLOS ONE. After careful consideration, we feel that it has merit but does not fully meet PLOS ONE’s publication criteria as it currently stands. Therefore, we invite you to submit a revised version of the manuscript that addresses the points raised during the review process.

We would appreciate receiving your revised manuscript by Dec 13 2019 11:59PM. To enhance the reproducibility of your results, we recommend that if applicable you deposit your laboratory protocols in protocols.io, where a protocol can be assigned its own identifier (DOI) such that it can be cited independently in the future. For instructions see: http://journals.plos.org/plosone/s/submission-guidelines#loc-laboratory-protocols

We look forward to receiving your revised manuscript.

Kind regards,

Rohina Joshi

Academic Editor

PLOS ONE

Journal Requirements

Additional Editor Comments (if provided):

Reviewers' comments:

Reviewer's Responses to Questions

**Comments to the Author**

1. Is the manuscript technically sound, and do the data support the conclusions?

Reviewer #1: Yes

Reviewer #2: Partly

2. Has the statistical analysis been performed appropriately and rigorously? 

Reviewer #1: N/A

Reviewer #2: N/A

3. Have the authors made all data underlying the findings in their manuscript fully available?

Reviewer #1: Yes

Reviewer #2: Yes

4. Is the manuscript presented in an intelligible fashion and written in standard English?

Reviewer #1: Yes

Reviewer #2: Yes

5. Review Comments to the Author

Reviewer #1: Specific comments

1. Page 4 line 54 – I believe the most up to date drowning figure from WHO is 360,000 deaths per year. Please check latest WHO data on drowning.

2. Page 4 line 56 – Please say upfront what the discrepancy is in this sentence – that WHO may be an underestimate

3. Page 4 lines 63 onwards – It is not clear from this introduction if you will be focussing on suicidal only, or also assaults? What do we know about assaults v suicide in terms of proportion of intentional drownings? Would be good to include to make it clear what the scope is, and that it’s important to consider both types of intentional drownings.

4. Page 5 line 77 – Please capitalise report titles

5. Page 5 line 100 – The question is ‘What are the risk factors’ rather than just identifying if that evidence for risk factors exists

6. Page 5 line 99 - Also, the proportion being investigated is the proportion of drowning as a cause for suicide, rather than the proportion of total drowning that are intentional. Please make this clear.

7. Page 6 line 107 – What do you mean exactly by the most exhaustive?

8. Table 2 – As per the aim in the Abstract, the objective is identify risk factors and mortality rates. In that case, was one of your inclusion criteria that the article needed to have calculated or assessed at least one of these in relation to intentional drowning?

9. Table 2 – If the article discussed both unintentional and intentional drowning, what was done?

10. Line 122 Page 7 – To be clear, how was the ‘substantiality’ assessment of the article done, what does this mean? Was this done with the McMaster Guideline as well? The sentence structure makes it appear substantiality and methodological quality are two different assessments you made. If not, please provide more information on what substantiality means and how it was assessed, otherwise please restructure the sentences.

11. Page 12 – Please cite Donaldson with year in-text and without the first name

12. Page 23 – Perhaps another limitation may be the timelines of the studies – bulk of which are almost 10 years old or older. The need for up to date research may be highlighted.

13. Would be useful to make a statement somewhere on the consistency of findings between the included studies – e.g. whether similar mortality rates / suicide proportion / risk factors have been found, and what this says about the quality and types of data available and need for further research.

Overall writing comments

- Please read through the document and ensure that the sentences are shorter. For example lines 57-61 on page 4 is cumbersome to read. Some other sentences are like this. Better to split these into separate sentences. Another such confusing sentence is on Page 15 line 11 onwards.

- Discussion was very well thought out and analysed. It can be made more succinct by grouping together the studies you have discussed that have similar findings.

Reviewer #2: Review

The manuscript is a systematic review of intentional drowning in Australia of rates and risk factors. The review includes 10 studies.

Abstract

The methods in the abstract say 2007 to 2018, but the results report trend on 1998- 2012 based on 8 studies. Am a bit confused- given the inclusion criteria findings should be limited to defined period.

Given the limited of studies and the details of methods of included study- perhaps it is more appropriate to call it a narrative synthesis. Risk factors are fundamentally demographic descriptive distribution of intentional drowning deaths.

Introduction

Minor

Line 67- uses both singular and plural, please revise

Line 74- uses “suicide- suicidal” twice. Using self- harm would read better.

Line 86- Except for Reference 9- the other reference are for suicide and psychiatric disorder in general, and not for drowning. The sentence is misleading in that it implies “suicide by drowning” is a unique case.

Line 96- Title states “fatal” so epidemiology would be for mortality- “particularly…” is perhaps not needed.

Second point in aims, line 100- “Is there any evidence available on the fatal intentional drowning risk factors in Australia?” is more suitable for evidence gap maps rather than SR. In my understanding from the abstract the abstract reports on risk factors for intentional drowning.

Results

Line 148- reference 54- based on the study objective of Ref 54 and the inclusion criteria mentioned in Table 2; this Is not a primary study in my opinion should not be included in the SR. Can the authors please explain a bit- on the rationale for including this study.

Page 12 does not have line numbers. The first paragraph again reports on findings beyond the study period

It is hard to take away anything from the analysis section

Table 5 is not particularly helpful- and repetitive on text.

Page 15 line 7 & 8- is the risk being compared for drowning suicide or only suicide?

Discussion - Table 6- does the journal accept table in discussion. Is repetitive of text above the table.

Discussion is fair bit repetition of the results. The authors refer to need for different types of intervention for intentional vs unintentional drowning deaths in the intro but do not come back / reflect on the same in the discussion. Presumably there is no study or data for indigenous populations- perhaps discuss why?

6. PLOS authors have the option to publish the peer review history of their article (what does this mean?). If published, this will include your full peer review and any attached files.

Reviewer #1: No

Reviewer #2: No

---

## [Author Response · Author response to Decision Letter 0]

12 Nov 2019

Thank you for the opportunity to revise our manuscript. We have addressed the reviewer's and editor's comments in the response to reviewers file we have uploaded.

---

## [Decision Letter · Decision Letter 1]

20 Feb 2020

PONE-D-19-21766R1

Fatal intentional drowning in Australia: A systematic literature review of rates and risk factors

PLOS ONE

Dear A/Prof. Franklin,

Thank you for submitting your manuscript to PLOS ONE. After careful consideration, we feel that it has merit but does not fully meet PLOS ONE’s publication criteria as it currently stands. Therefore, we invite you to submit a revised version of the manuscript that addresses the points raised during the review process.

We would appreciate receiving your revised manuscript by Apr 05 2020 11:59PM. To enhance the reproducibility of your results, we recommend that if applicable you deposit your laboratory protocols in protocols.io, where a protocol can be assigned its own identifier (DOI) such that it can be cited independently in the future. For instructions see: http://journals.plos.org/plosone/s/submission-guidelines#loc-laboratory-protocols

We look forward to receiving your revised manuscript.

Kind regards,

Laura Schwab-Reese

Academic Editor

PLOS ONE

Reviewers' comments:

Reviewer's Responses to Questions

**Comments to the Author**

1. If the authors have adequately addressed your comments raised in a previous round of review and you feel that this manuscript is now acceptable for publication, you may indicate that here to bypass the “Comments to the Author” section, enter your conflict of interest statement in the “Confidential to Editor” section, and submit your "Accept" recommendation.

Reviewer #1: (No Response)

2. Is the manuscript technically sound, and do the data support the conclusions?

Reviewer #1: Yes

3. Has the statistical analysis been performed appropriately and rigorously? 

Reviewer #1: I Don't Know

4. Have the authors made all data underlying the findings in their manuscript fully available?

Reviewer #1: Yes

5. Is the manuscript presented in an intelligible fashion and written in standard English?

Reviewer #1: Yes

6. Review Comments to the Author

Reviewer #1: This paper has markedly improved and is much clearer in its aims and outcomes. Please see comments below:

Pg 2, 28 1907 does not come in your range of 2007 to 2012 – please clarify that this is data and not the paper itself

Pg 2, 35 Please include the finding on what proportion of suicide deaths are from drowning – this is an important finding and of interest given the focus on suicide.

Pg 4, 55 LMIC context is not important to mention as Australia is not LMIC

Pg 4, 58-62 Can be summarised more succinctly, and focus more on the Australian studies. Otherwise, justify why you are talking about these countries in detail in particular and why a comparison is appropriate and useful.

Pg 4, 73 Would not say conflicting – there are varying proportions

Pg 4, 71-72 Am not sure in what population you are talking about for this finding

Pg 5, 84-85 Why exactly is it important to identify this? Please elaborate a little.

Pg 7, 124 Please specify that exclusion criteria is ‘Only unintentional drowning’

Pg 8, 142 The parentheses are a little unclear, I had to read it a few times to understand what you meant. Please restructure the sentence.

Pg 12 Can you please provide detail on how you used the results from papers looking at just one state, such as NSW or Qld. How were these rates converted and included in your national rate calculation?

Pg 16, 35 You have spent a lot of space discussing other high income countries – what is the conclusion you want the reader to come to? Please tell us the ‘so what’.

Pg 17, 51 Are there any theories as to why this proportion is lower in Australia? This seems a little counter intuitive given Australian’s better access to water bodies.

Pg 17, 58 Whilst the discussion on risk factors is interesting, it could perhaps be shortened. As these are only based on one study for each of the risk factors, there are no real additional learnings from the systematic review on this paper’s findings – you are reiterating what is already in the original paper. Where there is additional knowledge to be gained from bringing together more than one paper, this should be elaborated upon.

Pg 22, 175 If there are 10 papers, 67% can’t be the percentage.

Pg 22, 183-185 This sentence is unclear – not sure what you mean

General comments

- English needs polishing – there are several sentences that don’t make much sense on first reading. Would suggest reading it out loud before re-submitting.

- Figures are very unclear to read – not sure if this will be the case in finally online version

- There should be some acknowledgement that overall this type of suicide is not a major issue. I would suggest you comment on how specific interventions to target this exact type of suicide may not be a priority, but if common risk factors/ precursors appear with other suicide types, then intentional drownings can also be prevented along with other types of suicides.

7. PLOS authors have the option to publish the peer review history of their article (what does this mean?). If published, this will include your full peer review and any attached files.

Reviewer #1: No

---

## [Author Response · Author response to Decision Letter 1]

1 Apr 2020

Thank you to the reviewer for their ongoing feedback to improve our paper. We have uploaded a point by point response to the reviewer's feedback on the attach files page and this is included in the files submitted for this revision. Please view that file. We have uploaded higher res image files however the system appears to compress them when creating the proof. If the paper is accepted we will provide even higher resolution image files prior to publication.

---

## [Editor Report · Decision Letter 2]

3 Apr 2020

Fatal intentional drowning in Australia: A systematic literature review of rates and risk factors

PONE-D-19-21766R2

Dear Dr. Franklin,

We are pleased to inform you that your manuscript has been judged scientifically suitable for publication and will be formally accepted for publication once it complies with all outstanding technical requirements.

With kind regards,

Laura Schwab-Reese

Academic Editor

PLOS ONE
---

## [Editor Report · Acceptance letter]

13 May 2020

PONE-D-19-21766R2 

Fatal intentional drowning in Australia: A systematic literature review of rates and risk factors 

Dear Dr. Franklin:

I am pleased to inform you that your manuscript has been deemed suitable for publication in PLOS ONE. Congratulations! Your manuscript is now with our production department. 

With kind regards,

on behalf of

Dr. Laura Schwab-Reese 

Academic Editor

PLOS ONE